Journal of
Composites Science

MDPI

*Article*

# First-Principles Calculations of Crystallographic and Electronic Structural Properties of Au-Cu Alloys

**Dung Nguyen Trong [1,2,*], Van Cao Long [1], Umut Saraç [3], Van Duong Quoc [2] and Ştefan Ţălu [4,*]**

1 Institute of Physics, University of Zielona Góra, Prof. Szafrana 4a, 65-516 Zielona Góra, Poland
2 Faculty of Physics, Hanoi National University of Education, 136 Xuan Thuy, Cau Giay, Hanoi 100000, Vietnam
3 Department of Science Education, Bartın University, Bartın 74100, Turkey
4 The Directorate of Research, Development and Innovation Management (DMCDI), Technical University of Cluj-Napoca, 15 Constantin Daicoviciu St., 400020 Cluj-Napoca, Romania
* Correspondence: dungntdt2018@gmail.com (D.N.T.); stefan_ta@yahoo.com or stefan.talu@auto.utcluj.ro (Ș.Ț.)

**Abstract:** In this research, we have explored the effect of Au:Cu ratio on the crystallographic and electronic structural properties, formation energies, and radial distribution function (RDF) of Au-Cu alloy materials via density functional calculations. The results show that Au-Cu alloy can be formed in any Au:Cu ratio from 3:1 to 1:3 with a similar possibility. The results also reveal that the lattice constants of both Au and Cu are affected by the LDA-PWC pseudo-field, which is in full agreement with the experimental findings. An increase in the concentration of Cu impurity in Au results in a decrement not only in the lattice constants of the crystal system but also in the total energy of the system ($E_{tot}$). However, an enhancement in the electron density is determined by increasing Cu impurity concentration in Au. The RDF results confirm the contraction of lattice constants and a structural change in Au-Cu from cubic to tetrahedral is found when the Au:Cu ratio is equal to 1:1. These findings revealed in this work are expected to contribute to future studies on electronic materials.

**Keywords:** Au-Cu; DFT; electronic structures; formation energies; lattice system

check for updates

**Citation:** Trong, D.N.; Long, V.C.; Saraç, U.; Quoc, V.D.; Ţălu, Ş. First-Principles Calculations of Crystallographic and Electronic Structural Properties of Au-Cu Alloys. *J. Compos. Sci.* **2022**, *6*, 383. https://doi.org/10.3390/jcs6120383

Academic Editor: Prashanth Konda Gokuldoss

Received: 29 September 2022
Accepted: 8 December 2022
Published: 12 December 2022



## 1. Introduction

Today, with the quick development of science and technology, material science plays a very important, decisive role in the development and application of all scientific and technical disciplines. In particular, alloys are materials that are composed of two or more metals such as NiCu [1–3], FeC [4–6], NiFe [7,8], AlNi [9], NiAu [10], CuAu [11,12], WSi [13], FeCoNi [14], Al [15], Ni [16], etc. Alloys have great applicability in fields such as biology [17], adsorption [18], data storing [19], high density storage [20], photocatalysis [21], chemical sensors [22], and the biomedical field [23].

Among them, gold–copper (Au-Cu) alloys have been the subject of great interest to researchers because of their temperature-induced order–disorder transitions or long-period thermal stable structures [24–26]. It is well known that Au-Cu exists in three intermetallic phases, Au3Cu AuCu, and Cu3Au [25,27,28]. Au has a face centered cubic (FCC) system with a lattice constant a = 4.08 Å and a valence electron configuration $5d^{10}6s^1$, while Cu possesses an FCC system with a lattice constant a = 3.61 Å and a valence electron configuration $3d^{10}4s^1$. Au-Cu alloys exhibit excellent mechanical strength and chemical stability, high electrical conductivity, high resistance to corrosion, and high thermal conductivity [24,25,28]. Au-Cu alloys have many industrial and technological applications in various areas (such as communication [29], renewable energy [30], molecular sensing [31], medical imaging [32], and catalytic activities [28,33,34]).

Based on the studies published previously in the literature, one can emphasize that for Au-Cu alloys with tunable structures and their compositions and properties there are many experimental studies [35–38], as well as theoretical ones [26,33,34,39–44]. Recent experimental studies have shown that the Au:Cu mass ratio is one of the most important

factors affecting the photocatalytic properties, namely the best photocatalytic activity that is found for the Au-Cu alloys with Au:Cu mass ratio equal to 1:1 [38]. The theoretical studies on Au-Cu alloy can be performed using molecular dynamics [26,33,45], first-principle theory [39,41–44], and Monte Carlo simulation [27]. Most of these studies concentrate on Au-Cu nanoparticles or clusters.

Recently, scientists have investigated the effect of various factors on the Cu-Au alloy by molecular dynamics simulations [11]. In addition, scientists have examined the doping effect on the Au-Cu [12] and Ag-Au [46] alloy materials. The results of theoretical simulations show that there is a significant influence of doping concentration on alloys [12,46]. The cause of electronic structure formation in Ag-Au alloys has been revealed [46]. Here, a question arises, namely, what will happen when forming an electronic structure for Au-Cu alloy? The effect of Cu alloying in the Au lattice on the Au-Cu alloys, to our best knowledge, has not been studied to date, despite many existing studies, especially considering the formation processes of Au-Cu alloys. Hence, in this study, the density functional calculations have been performed using DMol3 tools in the framework of Materials Studio. The impact of Cu alloying on the crystallographic and electronic structural properties, formation energies, and radial distribution function (RDF) of Au-Cu alloy materials has been considered in detail.

## 2. Computational Methods

Initially, the Au-Cu alloy model was built within the framework of the Materials Studio software (Materials Studio is a proprietary software designed for materials simulation). The module package DMol3 [47] was used to calculate the electronic structure and band gap for the model and then the recovery and energy stability statistics were run in the software package. To determine the influencing factors of different correlation functions, the following approximations have been used: local density approximation (LDA) [48] which assumes that the exchange energy for the system at a point is equal to that of a homogeneous electron gas with the same density at that point (with two parametrizations of PWC [49] and VWN [50]), generalized gradient approximation (GGA) [51] which not only take the density at the point into account but also the gradient (change) of the density (with 3 parametrizations of PBE [52], RPBE [53], PW91 [49]), to perform geometry optimizations. To determine the electronic structure, the k-point grid pattern of the Monkhorst–Pack scheme [54] with the corresponding configurations of Au and Cu, respectively, $5d^{10}6s^1$ and $3d^{10}4s^1$, was used. The valence electron–core interactions are described by density functional semi-core pseudo-potential (DSPP) [55] with a change in total energy for $1 \times 10^{-6}$ eV.

## 3. Results and Discussion

### 3.1. Pseudo-Potential Selection

Figure 1 demonstrates the dependence of the optimized lattice constants of pure Au and Cu metals for different pseudo-potentials: GGA-PW91, GGA-PBE, GGA-RPBE, LDA-VWN, and LDA-PWC. The corresponding lattice constants of Au are a = b = c = 4.1724 Å, 4.1735 Å, 4.1924 Å, 4.0910 Å, and 4.0910 Å, whereas for Cu they are a = b = c = 3.6564 Å, 3.6564 Å, 3.6965 Å, 3.5710 Å, and 3.5710 Å. The lattice constants obtained by LDA-PWC and LDA-VWN are closest to the experimental values (4.0782 Å for Au and 3.6149 Å for Cu) [56] and simulation values [12], so LDA-PWC was selected as suitable for use for later calculations.

To study the effect of alloying, in each system of Au there are four atoms (Figure 2(a1)), so the numbers of Cu atoms (Figure 2(b1)) that can be substituted into the Au lattice are one, two, and three, which correspond to the 3:1, 1:1, and 1:3 Au:Cu atom ratios, respectively. For each Au:Cu ratio, two different models were created: with Au:Cu = 3:1, the two models were Au3Cu-c (Figure 2(a2)) and Au3Cu-f (Figure 2(b2)); with Au:Cu = 1:1, the two models were AuCu-cf (Figure 2(a3)) and AuCu-ff (Figure 2(b3)), and with Au:Cu = 1:3, the two models were AuCu3-cff (Figure 2(a4)) and AuCu3-fff (Figure 2(b4)). The difference between the two corresponding models was fixed by the position of the substituted Cu atom in

the Au lattice. All structures of Au-Cu alloys with different Au:Cu ratios are depicted in Figure 2.

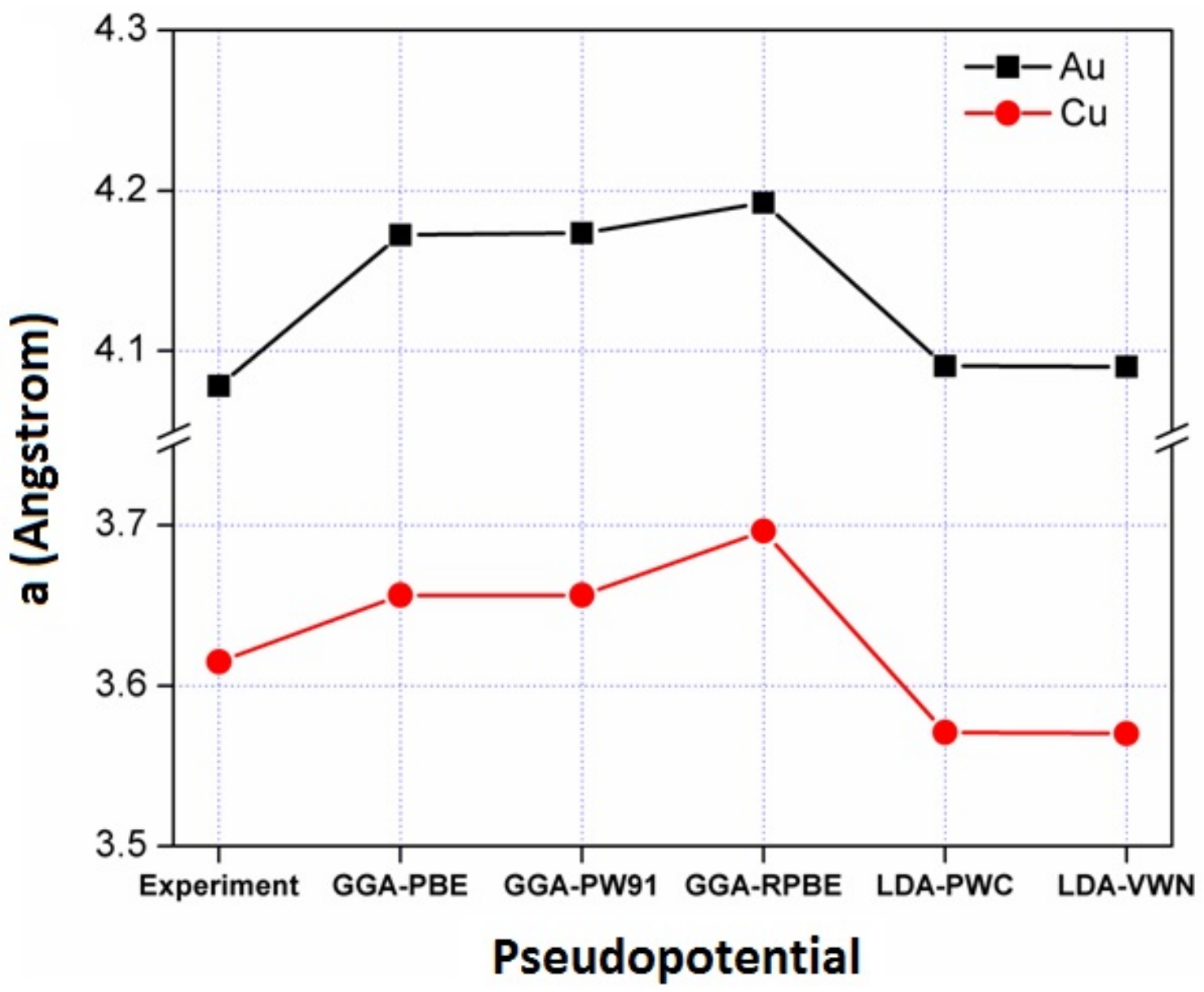


**Figure 1.** Optimization findings of Au and Cu using various pseudo-potentials.

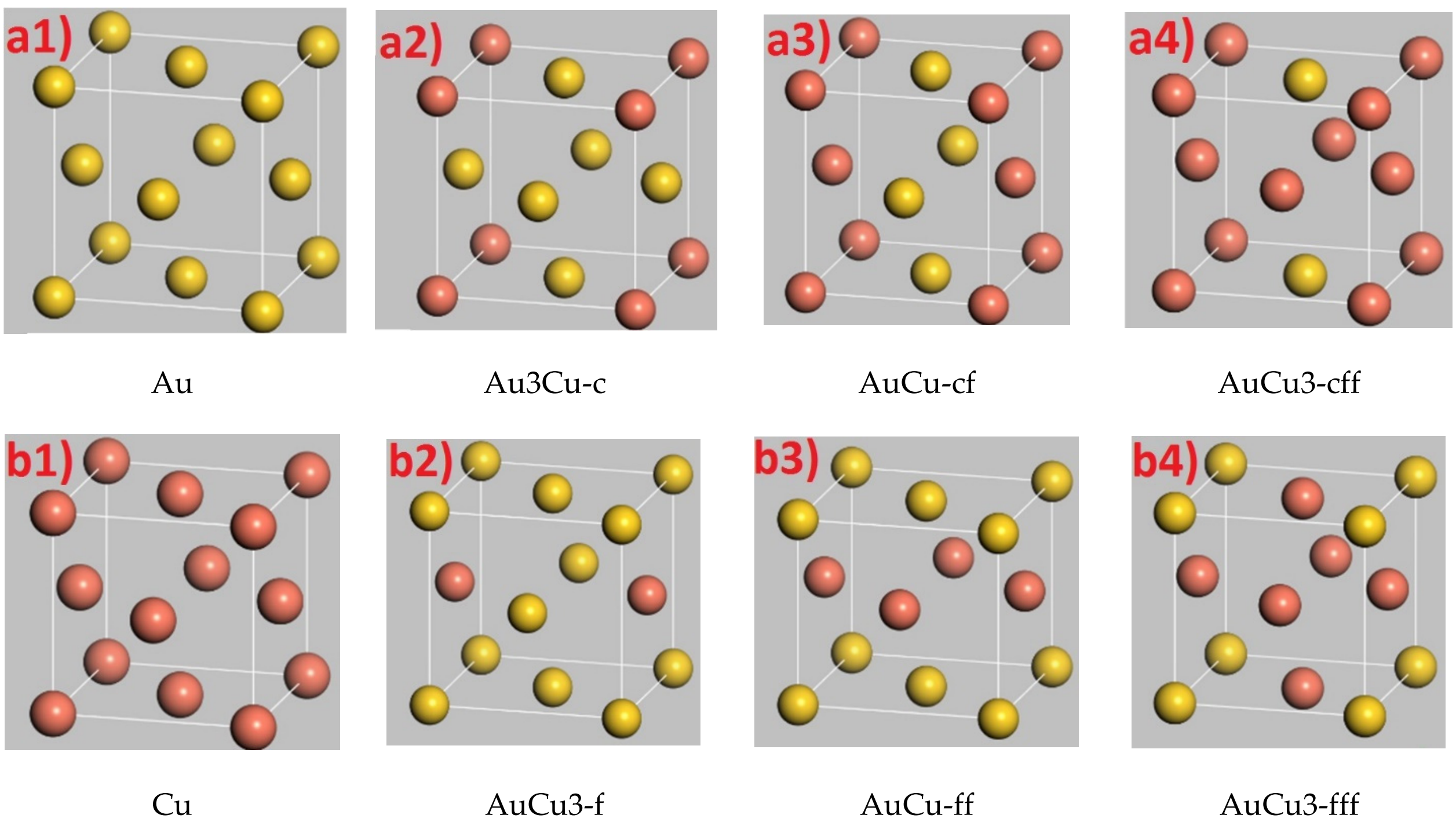


**Figure 2.** Optimized systems of Au, Au3Cu, AuCu, AuCu3, and Cu (yellow: Au, pink: Cu).

The optimized lattice constants and the total energies of systems of Au, Cu, and their alloys were determined using LDA parametrized by Perdew–Wang (LDA-PWC) [49] and are shown in Figure 3.

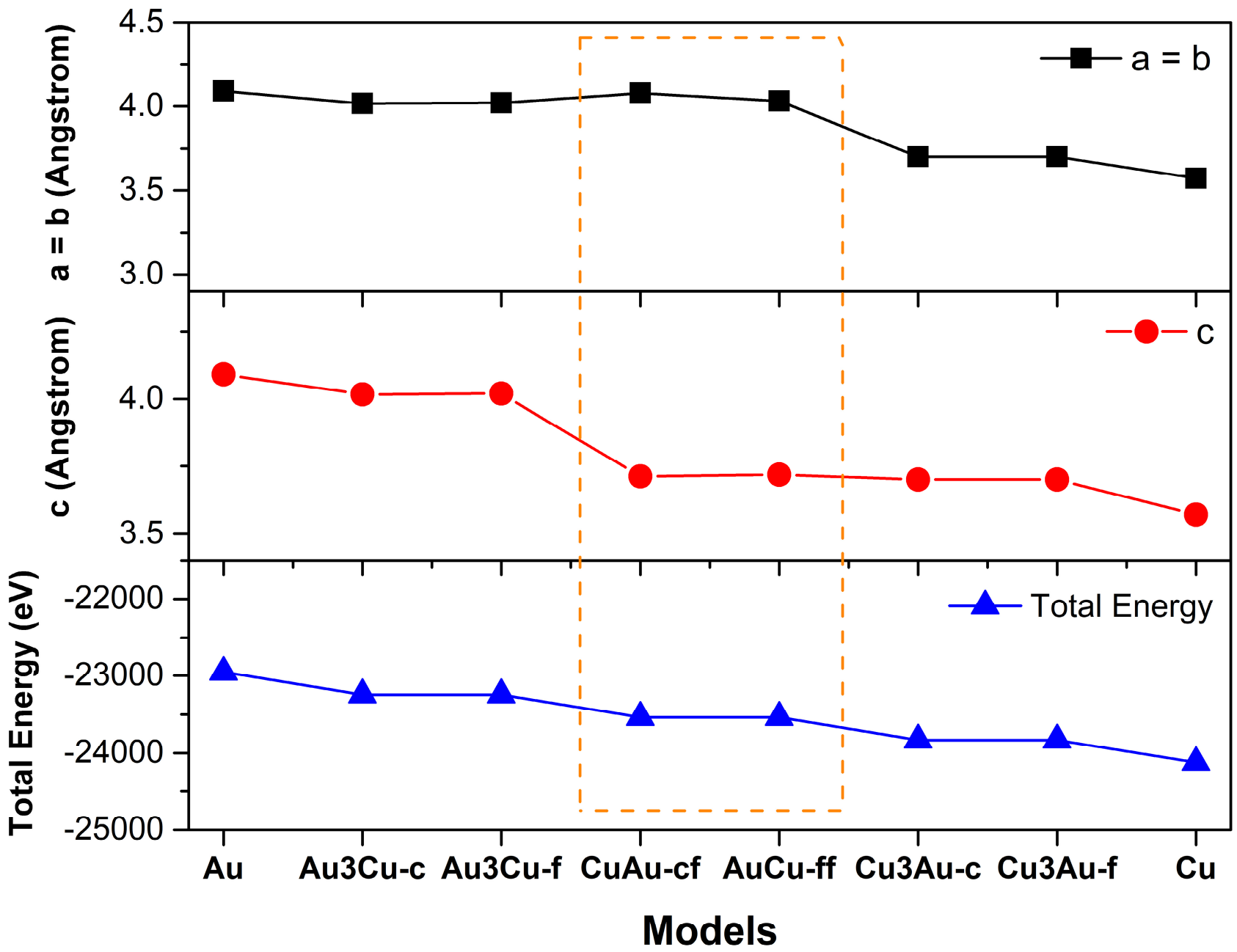


**Figure 3.** Optimized lattice constants and total energies for different systems of Au, Au3Cu, AuCu, AuCu3, and Cu materials were calculated using LDA-PWC.

The variation of lattice constants and total energies of the systems have been determined using LDA-PWC and are shown in Figure 3. The optimized systems of Au and Cu are a cubic system with the corresponding lattice constant a = b = c = 4.0910 Å with Au and 3.5710 Å with Cu, respectively. Table 1 shows that the optimized systems of Au3Cu-c and Au3Cu-f have an FCC structure with the same lattice constants a = b = c = 4.0170 Å, so the optimized structure is labeled as Au3Cu; the optimized systems of AuCu3-cff and AuCu3-fff have an FCC structure with a = b = c = 3.7002 Å, the optimized structure is called AuCu3. For AuCu-cf and AuCu-ff models, the optimized systems have a tetragonal structure with a = b = 4.0325 Å and c = 3.7185 Å, the optimized systems are called AuCu. The results obtained are in full agreement with simulation [12] and experimental [56] studies.

**Table 1.** Structural properties of Au, Au3Cu, AuCu, AuCu3, and Cu materials were calculated using LDA-PWC.

| | **Au** | **Au3Cu** | **AuCu** | **AuCu3** | **Cu** |
|---|---|---|---|---|---|
| Bravais Lattice | FCC | Triclinic | Triclinic | Triclinic | FCC |
| a = b (Å) | 4.0910 | 4.0170 | 3.7185 | 3.7002 | 3.5710 |
| c (Å) | 4.0910 | 4.0170 | 4.0325 | 3.7002 | 3.5710 |
| Total Energy $E_{tot}$ (eV) | −22,952 | −23,245 | −23,538 | −23,830 | −24,124 |

An increase in the Au:Cu ratio causes a decrement in the lattice constants of alloys because of the different atomic radii of Au (144 pm) and Cu (128 pm). The larger atoms of Au are replaced by smaller atoms of Cu, which reduces the systems of Au lattice and creates a smaller system of Au-Cu alloys. The smaller lattice constants than Au3Cu and AuCu3 show that the ratio of Au:Cu = 1:1 causes a periodic substitution of Cu atoms into Au lattices, and creates a periodic structure of Au and Cu atoms in space, as well as changes the lattice system of Au from cubic to tetragonal.

On the other hand, when building the Au:Cu = 1:1 model, the Au and Cu atoms in the base cell are assumed to be arranged in parallel planes. As the size of Cu ions is smaller, the distance between two planes containing Au atoms will be smaller than the

distance between a plane containing Au atoms and the plane containing Cu atoms next to it. Changing this distance will lead to a structural phase transition of Au:Cu.

Thus, the decrement in alloy lattice constants leads to two consequences: (i) the decrease in total energy of their unit cells and (ii) the change in the lattice system from cubic to tetragonal, as listed in Table 1.

### 3.2. Formation of Au-Cu Alloys

The process of forming Au-Cu alloy can be determined through the transformation process ($\Delta E_{mod}$) and is determined by Formula (1) [57]:

$$\Delta E_{mod} = E_{tot}\ (\text{models}) - E_{tot}\ (\text{Au}) + m \cdot \mu_{Au} - n \cdot \mu_{Cu}, \tag{1}$$

where $\mu_{Au}$ and $\mu_{Cu}$ are −5738 eV and −6031 eV and represent the chemical potentials of Au and Cu; $E_{tot}$ (Au) and $E_{tot}$ (model) are the sums of the energies of pure copper (Au) and the Au-Cu alloy model. In this formula, m represents the number of substituted Au atoms, while n corresponds to the removed Cu atoms in each model and they are illustrated in Table 2.

**Table 2.** m and n values for different models of Au-Cu alloys and their corresponding formation energies.

| Models | Au | Au3Cu | AuCu | AuCu3 | Cu |
|---|---|---|---|---|---|
| m | 0 | 1 | 2 | 3 | 4 |
| n | 0 | 1 | 2 | 3 | 4 |
| Formation Energy ($\times 10^{-2}$ eV) | 0 | 0.544 | 11.973 | 97.143 | 0 |

The formation energy of all Au-Cu models is positive and small. Furthermore, it is enhanced by increasing the Cu concentration. The small and positive formation energies suggest that Au-Cu alloys are difficult, but possible, to form with different Au:Cu ratios and the obtained materials are stable. The formation energies also show that all three Au-Cu alloys (Au3Cu, AuCu, and AuCu3) can be formed in the same process, or Au and Cu can be melted together in different ratios (3:1, 1:1, and 1:3) to create alloys. In general, it can be predicted that Au-Cu alloys can be formed in distinct mass or atom ratios.

### 3.3. Electronic Structures of Au, Cu, and Au-Cu Alloys

Figure 4 shows the partial density of states (PDOS) and band structure for Au, Cu, metals, and Au:Cu alloy determined in the band gap between −10 eV and 15 eV. The band structures and state densities of Au and Cu were calculated using LDA-PWC. The dotted line shows the Fermi level, corresponding to $E_g$ = 0 eV. The results of calculations show that the valance band (VB) has a high electron density but is narrower than the conduction band (CB), which is shown in Figure 4a,e. Both band structures and PDOSs of Au and Cu show the overlapping of CBs and VBs and here no band gap exists, which shows that Au and Cu are metals. The PDOSs of Au and Cu (Figure 4a,e) show the contributions of the orbitals to the electron structure. In it, the contribution of the s and p orbitals is small and dominated by the d orbitals. The result is in full agreement with their valence electrons ($5d^{10}6s^1$ and $3d^{10}4s^1$).

The corresponding band structures and total density of states (DOS) of Au3Cu (Figure 4b), AuCu (Figure 4c), and AuCu3 (Figure 4d) alloys are also demonstrated in Figure 4. In this figure, the dotted line shows the Fermi level at 0.0 eV. All these results show that the band structure of Au-Cu alloy is very similar to that of Au, Cu metals, and Au:Cu alloy. The energy level of the VBs of Au-Cu was determined in the range from −10 eV to 15 eV, which has a higher state density value than metal. These results are in accordance with the simulation [39] and experimental [48] findings.

In addition to the previously analyzed results, there are also results showing the mobility of conduction and valence electrons in the s, p, d, and f subshells. It follows that

in the s, p, and f subshells, the electron mobility in the two valence and conduction bands is almost unchanged and it only changes strongly in the d subshell, which confirms the metals. Au, Cu, and Au:Cu alloys, when doped, have highly active electrons in the d subshell. When changing the energy levels, the electron mobility of Au metal in the conduction band is greater than that of Cu and reaches the maximum value when the doping ratio of Cu to Au is 50% (AuCu), then a peak appears. The priority is near the energy level E = −2.5 eV and two small peaks with Au and Cu metals with lower heights show that with pure Au and Cu metals, there are two energy levels with mobility. The conduction band electrons have nearly equal values (according to the results shown in Figure 4c).

At room temperature (T), T = 300 K, almost all energy levels below Fermi levels were filled; at room temperature, the electrons jumped to the CBs to create free electrons, which leads to the thermal and electrical conductivity of Au-Cu alloys (higher compared to the Au and Cu metals).

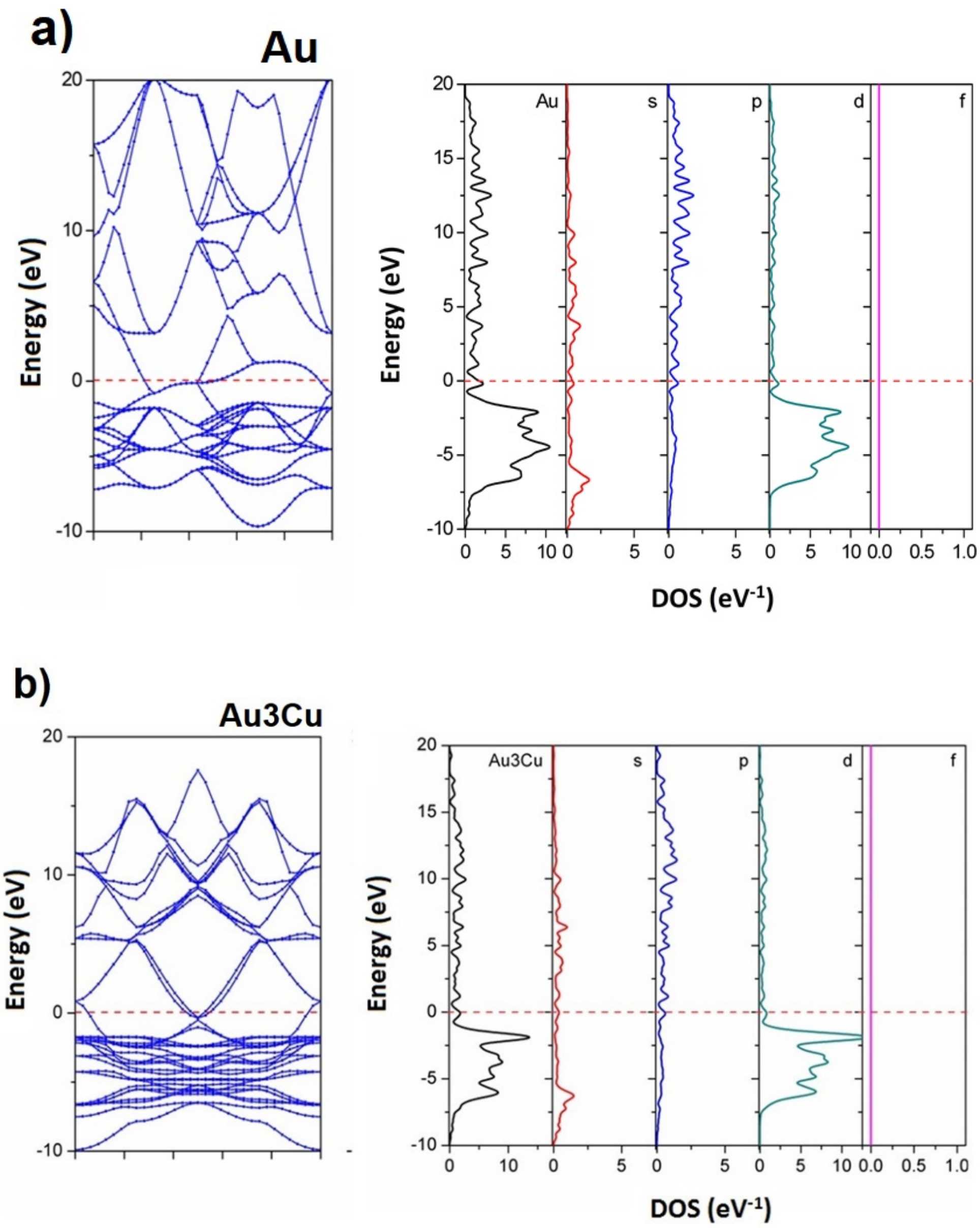


**Figure 4.** *Cont.*

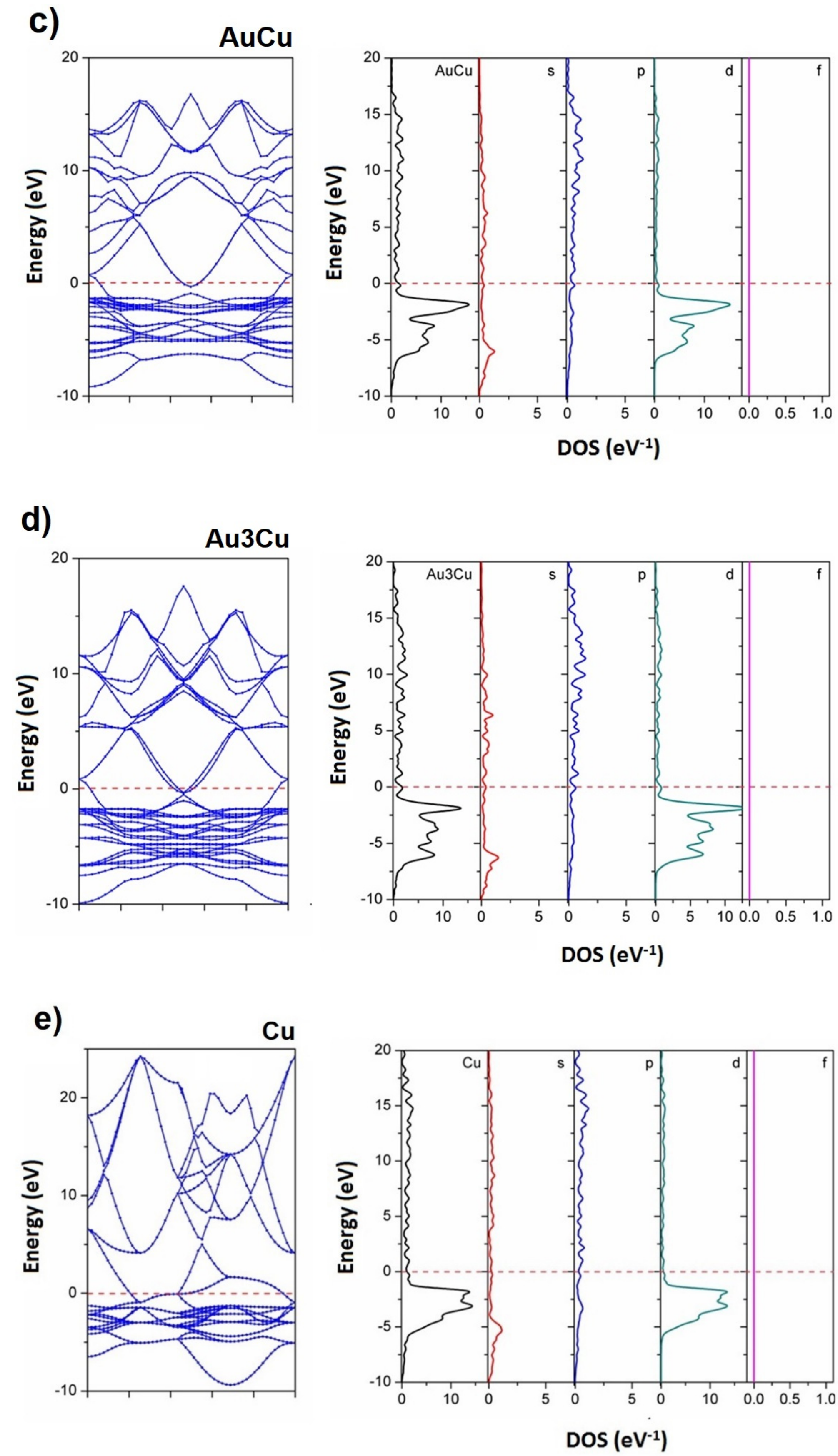


**Figure 4.** PDOSs and electronic structures of Au (**a**), Au3Cu (**b**), AuCu (**c**), AuCu3 (**d**), and Cu (**e**) were calculated using LDA-PWC.

The results of calculations show that the state density of Au changes hugely when it has doping with Cu (Figure 4). The DOSs of Au only have a wide zone, but the DOSs of Au-Cu alloys show two clear zones. In consequence, the mobility of electrons in valence bands is decreased when Cu is doped into the Au lattice. The calculated widths of VB and CB (given in Table 3) show that the width of VB of Au-Cu alloys is slightly decreased and then increased when Cu is doped into an Au lattice (the largest reduction happens when the doping concentration is 50%). Table 3 also shows that the doping of Au leads to a decrease in the width of CB. These results suggest that the electron mobility of Au-Cu is smaller than that of Au or Cu metals. The corresponding widths of VB and CB are shown in Table 3.

**Table 3.** Widths of VB and CB for different Au-Cu alloys models.

| Models | VB Widths | CB Widths |
|---|---|---|
| Au | 10.3945 | 23.2789 |
| Au3Cu | 10.3154 | 20.1977 |
| AuCu | 10.2024 | 21.1547 |
| Cu3Au | 11.0008 | 22.9690 |
| Cu | 10.3949 | 23.2789 |

### 3.4. RDF Analysis

The RDF g(r) of Au, Au3Cu, AuCu, Cu3Au, and Cu calculated using LDA-PWC are shown in Figure 5. The periodic lattices of Au, Au3Cu, AuCu, Cu3Au, and Cu lead to the appearance of separated peaks at different positions corresponding to various atom distances in the lattice of materials. Figure 5 shows that the corresponding peaks shift to the larger distance when the Au:Cu ratio decreases, which shows that the substitution of Cu atoms into the Au lattice leads to the expansion of lattice constants. At the second peak position (4.0910 Å to 3.5710 Å) in the RDF of AuCu-cf and AuCu-ff models, two separate peaks appear instead of only one for Au, Au3Cu, AuCu, Cu3Au, and Cu.

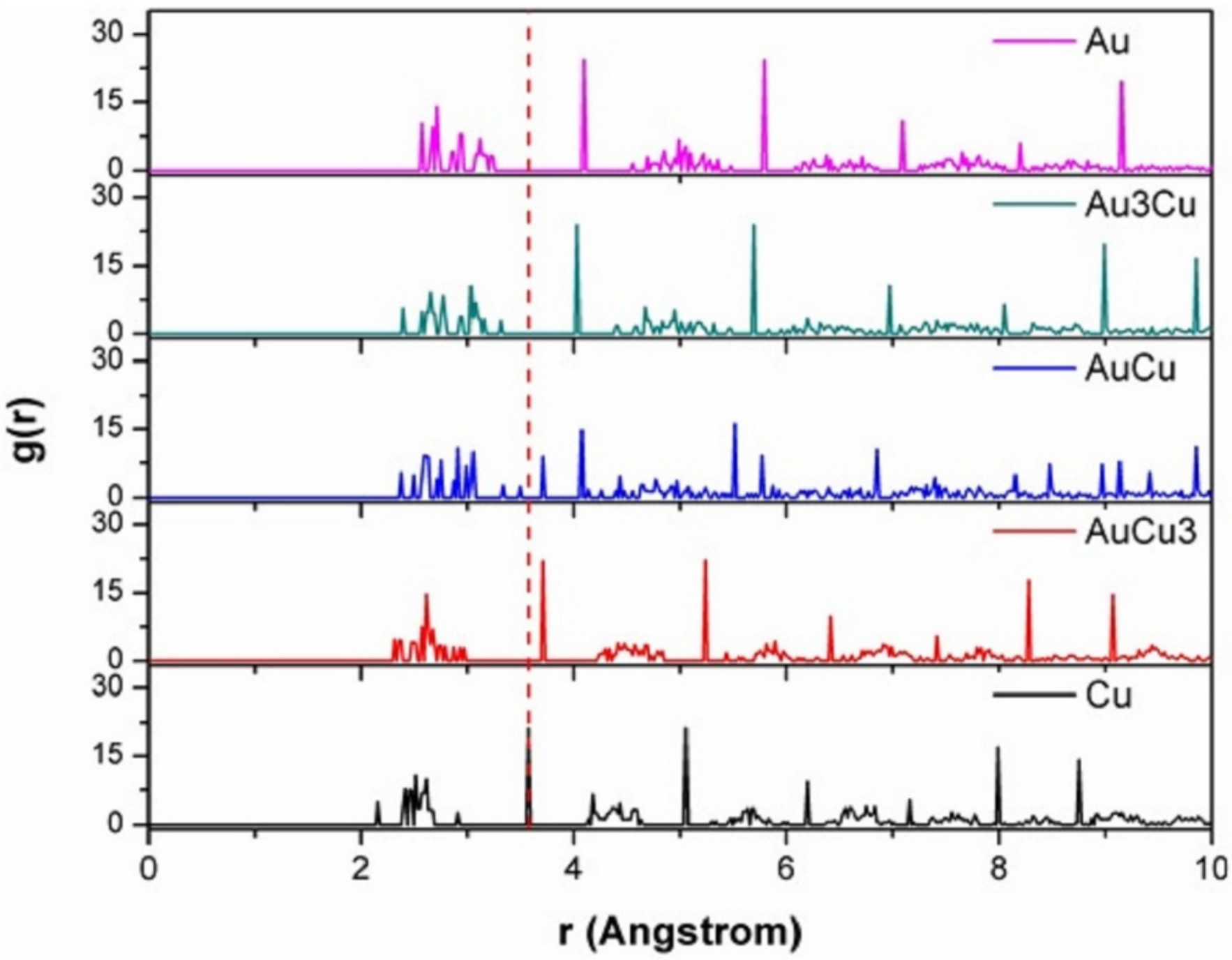


**Figure 5.** RDF g(r) of Au, Au3Cu, AuCu, AuCu3, and Cu was calculated using LDA-PWC.

The appearance of these two peaks suggests that AuCu has two different lattice constants, which can be explained by the structure of the systems of AuCu in Figure 2. For Au-Cu alloys with Au:Cu = 1:1, two periodic lattices of Au and Cu appeared in space with two lattice constants, which lead to the appearance of two peaks in RDF. Similar results can be obtained at the 6, 4, and later peak positions.

In Section 3.1 above, the influence of different pseudo-potentials on the lattice constant and energy of Au and Cu metals is determined in order to determine the type of pseudo-potential suitable for Au:Cu doped alloys. In Section 3.4, the RDF curves of each metal Au, Cu, and doped alloys Au3Cu, AuCu, AuCu3 were determined and, in consequence, the bond lengths between Au-Au, Au-Cu, and Cu-Cu were calculated. These results are completely different, one is the lattice constant of the base cell and the other is the bond length between atoms in a base cell.

The results represented above suggest that the substitution of Cu atoms into Au lattice leads to a decrease in the size of lattice constants.

## 4. Conclusions

In this research, the crystallographic and electronic structural properties, formation energies, and RDF of Au-Cu alloys with different Au:Cu ratios have been successfully investigated using density functional calculations in the framework of the DMol3 package. The results show that Au-Cu alloy can be formed in any Au:Cu ratio from 3:1 to 1:3 with a similar possibility. The results also show that the LDA-PWC pseudo-field affects the lattice constants of both Au and Cu. An increase in the concentration of Cu impurity in Au causes a decrement in the lattice constant (a = b = c) of Au-Cu alloy from 4.0910 Å to 3.5710 Å. The results also show a decrement in the $E_{tot}$ value of the system from −22,952 eV to −24,124 eV when increasing the Cu concentration. When the Au:Cu is 1:1, a structural change in Au-Cu from cubic to tetrahedral is detected. The determining widths of VB and CB suggest that the electron mobility of Au-Cu alloys increases when the Cu concentration increases. The RDF analysis confirms the contraction of the lattice constants by the appearance of new peaks. These findings revealed in this study are expected to contribute to future studies on electronic materials.

**Author Contributions:** D.N.T.: Conceptualization, Methodology, Investigation, Validation, Writing—original draft, Writing—review and editing, Formal analysis, Supervision. V.C.L.: Writing—original draft; Formal analysis. U.S.: Writing—original draft; Formal analysis, editing. V.D.Q.: Writing—original draft. Ş.Ţ.: Writing—review and editing. All authors have read and agreed to the published version of the manuscript.

**Funding:** This research received no external funding.

**Institutional Review Board Statement:** Not applicable.

**Informed Consent Statement:** Not applicable.

**Data Availability Statement:** The data that support the findings of this study are available from the corresponding author upon reasonable request.

**Conflicts of Interest:** The authors declare no conflict of interest.

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
