# Peer review of "First-Principles Calculations of Crystallographic and Electronic Structural Properties of Au-Cu Alloys"

_jcs, doi:10.3390/jcs6120383_

Round 1

Reviewer 1 Report

In this work, the authors reported the crystallographic and electronic structural properties, formation energies and RDF of Au-Cu alloys with different Au:Cu ratio using density functional calculations. The work is meaningful and interesting, but there is some issues should be addressed before publication:

1.     The calculated results indicate a decrement in the Etot value from -22952 eV to -24124 eV with increasing the Cu concentration. When the Au:Cu is 1:1, a structural change of Au-Cu from cubic to tetrahedral is also detected. The authors should explain the reason for these results.

2.     The authors carried out the calculation work according to the experimental result with Cu:Au ratio ranging from 1:3 to 3:1. How about the possibility of the formation of CuAu alloy with other molar ratio beyong.

3.     The language should be carefully polished, there are too many mistakes and errors. Just in the introduction section as example, Among them, alloys such as NiCu 32 [1-3], FeC [4-6], NiFe [7-8], AlNi [9], NiAu [10], CuAu [11, 12], WSi [13], FeCoNi [14], Alloys are materials that are composed of 2 or more metals such as Al [15], Ni [16], ..

Author Response

Review 1

In this work, the authors reported the crystallographic and electronic structural properties, formation energies and RDF of Au-Cu alloys with different Au:Cu ratio using density functional calculations. The work is meaningful and interesting, but there is some issues should be addressed before publication:

Answer:

The authors have appreciated all comments and suggestions on the structure of our manuscript.

1.The calculated results indicate a decrement in the Etot value from -22952 eV to -24124 eV with increasing the Cu concentration. When the Au:Cu is 1:1, a structural change of Au-Cu from cubic to tetrahedral is also detected. The authors should explain the reason for these results.

Answer:

When the Cu doping concentration is changed - since the radius of Cu ions (128 pm) is smaller than that of Au (144 pm) - the lattice constant of the alloy will decrease, and the total energy of the base cell decreases. On the other hand, when building the Au:Cu=1:1 model, the Au and Cu atoms in the base cell are assumed to be arranged in parallel planes. Because the size of Cu ions is smaller, the distance between two planes containing Au atoms will be smaller than the distance between a plane containing Au atoms and the plane containing Cu atoms adjacent to it. Changing this distance will lead to a structural phase transition of Au:Cu.

2.The authors carried out the calculation work according to the experimental result with Cu:Au ratio ranging from 1:3 to 3:1. How about the possibility of the formation of CuAu alloy with other molar ratio beyong.

Answer:

In this paper, the authors only perform calculations for the Cu:Au ratios ranging from 1:3 to 3:1 because the construction of these structures only requires a small base cell, which is convenient for analysis in using low profile computers. Models with other ratios (less than 1:3) or greater than 3:1 can be built and calculated, but this has certain limitations. Reducing the scale will not match the reality of the alloys, and will increase the base cell size in calculation. Also, increasing the base cell will lead to the need to use powerful computer systems that are not suitable for the research conditions of the author's group.

3.The language should be carefully polished, there are too many mistakes and errors. Just in the introduction section as example, Among them, alloys such as NiCu 32 [1-3], FeC [4-6], NiFe [7-8], AlNi [9], NiAu [10], CuAu [11, 12], WSi [13], FeCoNi [14], Alloys are materials that are composed of 2 or more metals such as Al [15], Ni [16], ..

Answer:

The author has revised the entire style, language, and grammatical errors in English, hoping to meet the requirements of the reviewer.

Reviewer 2 Report

1.       In section 2, the author listed different correlation functions, please explain the differences between these correlation functions in the manuscript.

2.       In section 3.1the lattice constants obtained by LDA-PWC and LDA-VWN are closest to the experimental valuesplease explain why only the LDA-PWC was elected to use for later calculations

3.       In section 3.1, the author used different pseudopotentials to calculate the lattice constants of Au and Cu, please explain the reason for the difference in lattice constants.

4.       The author mentioned in 3.1 “An increase in the Au:Cu ratio causes a decrement in the lattice constants of alloys”, please explain in the manuscript how the decrement of the lattice constant inpacted the properties of the Au-Cu compounds.

5.       In section 3.3, the author should conduct an in-depth analysis of PDOSs and band structure. As we know, the result you got from the simulation calculation was what we all have already knownsuch as Au and Cu are undoubtly metal, so any more helpful conclusions

6.       In section3.4, which peak of RDF curve is the lattice constant related to? And in section 3.1, the author had calculated the lattice constants, what is the role of RDF analysis?

7.       Authors have mentioned many times that your conclutions are in accordance with othershoweverthere is almost nothing special about your own research significanceplease add some corresponding statement.

8.       English in the manuscript should be improved. For example, the sentence “The results obtained indicate that there is a great influence of doping concentration.” Two predicate verbs in one sentence at the same time. There are many similar problems in the manuscript.

Author Response

Review 2

In section 2, the author listed different correlation functions, please explain the differences between these correlation functions in the manuscript.

Answer:

The authors have updated the content in the manuscripts to explain the difference between these correlation functions.

2.In section 3.1,the lattice constants obtained by LDA-PWC and LDA-VWN are closest to the experimental values,please explain why only the LDA-PWC was elected to use for later calculations ?

Answer:

In this paper, the authors choose only one approximation of LDA-PWC for calculation for the following reasons: (i) LDA-PWC and LDA-VWN both give the lattice constants of Au and Cu close to the experimental one. Most importantly, (ii) the total energy of the Au and Cu base cells after optimization is also calculated through LDA-PWC and PDA-VWN. The calculation results using LDA-PWC for total energy are smaller, which means that the base cell is more durable. From those results, the authors only selected LDA-PWC to use without using LDA-VWN.

3.In section 3.1, the author used different pseudopotentials to calculate the lattice constants of Au and Cu, please explain the reason for the difference in lattice constants.

Answer:

The reason for the difference in lattice constants with different pseudopotential fields is that for each different type of pseudopotential field, the interaction between different electrons will be studied, leading to the lattice constant and the energy of the system will be different. We have added in the content of the manuscript to clarify this.

  1. The author mentioned in 3.1 “An increase in the Au:Cu ratio causes a decrement in the lattice constants of alloys”, please explain in the manuscripthow the decrement of the lattice constant inpacted the properties of the Au-Cu compounds.

Answer:

The authors have updated the manuscripts to explain the effects of lattice constant decrement on the properties of Au-Cu alloys.

5.In section 3.3, the author should conduct an in-depth analysis of PDOSs and band structure. As we know, the result you got from the simulation calculation was what we all have already known, such as Au and Cu are undoubtly metal, so any more helpful conclusions ?

Answer:

In addition to the previously analyzed results, there are also results showing the mobility of conduction and valence electrons in the s, p, d, and f subshells. The results show that in the s, p, and f subshells, the electron mobility in the two valence and conduction bands is almost unchanged and only changes strongly in the d subshell, which confirms the metals. Au, Cu and Au:Cu alloys, when doped, have highly active electrons in the d subshell. When changing the energy levels, the electron mobility of Au metal in the conduction band is larger than that of Cu and reaches the maximum value when the doping ratio of Cu to Au is 50% (AuCu), a peak appears. The priority is near the energy level E=-2.5eV and two small peaks with Au and Cu metals with lower heights appear showing that with pure Au and Cu metals, there are two energy levels with mobility. The conduction band electrons have nearly equal values, the results are shown in Figure 4c.

6.In section3.4, which peak of RDF curve is the lattice constant related to? And in section 3.1, the author had calculated the lattice constants, what is the role of RDF analysis?

Answer:

In section 3.1, the authors determined the influence of different pseudopotentials on the lattice constant and energy of Au and Cu metals in order to determine the type of pseudopotential suitable for Au:Cu doped alloys. In section 3.4, the author determines the RDF curves of each metal Au, Cu and doped alloys Au3Cu, AuCu, AuCu3 to determine the bond lengths between Au-Au, Au-Cu, Cu-Cu. These two results obtained are of completely different nature, one is the lattice constant of the base cell and the other is the bond length between atoms in a base cell.

7.Authors have mentioned many times that your conclusions are in accordance with others,however,there is almost nothing special about your own research significance,please add some corresponding statement.

Answer:

The author has edited, supplemented and explained the results in accordance with the previous results and added the latest conclusions to the content of the manuscript.

8.English in the manuscript should be improved. For example, the sentence “The results obtained indicate that there is a great influence of doping concentration.” Two predicate verbs in one sentence at the same time. There are many similar problems in the manuscript.

Answer:

The author has revised the entire English grammar and style, hopefully with this revision, it has met the requirements of the reviewer. The sentence given by Reviewer has been corrected, similarly in other places of the text.

Round 2

Reviewer 1 Report

it is now suitable for publication.

Reviewer 2 Report

  • The author has modified it according to the requirements, it is recommended to accept.